# An abiotic source of Archean hydrogen peroxide and oxygen that pre-dates oxygenic photosynthesis

Hongping He [1,2,3,6✉], Xiao Wu [1,2,3,6], Haiyang Xian [1,2], Jianxi Zhu [1,2,3], Yiping Yang[1,2,3], Ying Lv [1,2,3], Yiliang Li [4✉] & Kurt O. Konhauser[5✉]

The evolution of oxygenic photosynthesis is a pivotal event in Earth's history because the $O_2$ released fundamentally changed the planet's redox state and facilitated the emergence of multicellular life. An intriguing hypothesis proposes that hydrogen peroxide ($H_2O_2$) once acted as the electron donor prior to the evolution of oxygenic photosynthesis, but its abundance during the Archean would have been limited. Here, we report a previously unrecognized abiotic pathway for Archean $H_2O_2$ production that involves the abrasion of quartz surfaces and the subsequent generation of surface-bound radicals that can efficiently oxidize $H_2O$ to $H_2O_2$ and $O_2$. We propose that in turbulent subaqueous environments, such as rivers, estuaries and deltas, this process could have provided a sufficient $H_2O_2$ source that led to the generation of biogenic $O_2$, creating an evolutionary impetus for the origin of oxygenic photosynthesis.

[1] CAS Key Laboratory of Mineralogy and Metallogeny/Guangdong Provincial Key Laboratory of Mineral Physics and Materials, Guangzhou Institute of Geochemistry, Chinese Academy of Sciences, 510640 Guangzhou, China. [2] CAS Center for Excellence in Deep Earth Science, 510640 Guangzhou, China. [3] University of Chinese Academy of Sciences, 100049 Beijing, China. [4] Department of Earth Sciences, The University of Hong Kong, 999077 Hong Kong, China. [5] Department of Earth and Atmospheric Sciences, University of Alberta, Edmonton, Alberta T6G 2E3, Canada. [6] These authors contributed equally: Hongping He, Xiao Wu. ✉email: hehp@gig.ac.cn; yiliang@hku.hk; kurtk@ualberta.ca

The evolution of oxygenic photosynthesis was a critical biological innovation that allowed water to be used as an electron source and dioxygen gas ($O_2$) to be released as a metabolic by-product to the early atmosphere. Oxygenic photosynthesis operates by a four-electron reaction (R1) process associated with chlorophyll-$a$ and the water-oxidizing complex (WOC)[1,2]. However, it would have been difficult for the ancestor of modern cyanobacteria to extract four electrons from water, a very stable compound, before the development of chlorophyll-$a$ and the WOC[2–4]. Thus, it has been proposed that there might have been a transitional electron donor prior to $H_2O$ for the evolution of oxygenic photosynthesis on the early Earth[3].

$$2H_2O \rightarrow O_2 + 4H^+ + 4e^- \qquad (1)$$

Many intermediates have been proposed[1,3,5–8], but only hydrogen peroxide ($H_2O_2$), divalent manganese ($Mn^{2+}$), and bicarbonate ($HCO_3^-$) satisfy the multi-electron chemistry requirements. As an early intermediate candidate prior to $H_2O$, $H_2O_2$ is plausible for two reasons. First, it can be oxidized at electrochemical potentials that are accessible to existing anoxygenic phototrophs[1], meaning that if photosynthesis evolved as an anoxygenic process, as is generally accepted[9], the photosynthetic machinery might already have been in place to handle $H_2O_2$ redox transformations. Second, $H_2O_2$ and the by-product ($O_2$) would have consumed the surrounding reductants used by anoxygenic photosynthesizers (e.g., $H_2$, $H_2S$, $Fe^{2+}$), creating evolutionary pressure that forced existing photoautotrophs to adapt to locally oxidized environments and to utilize new electron donors[6], assuming a large sustained $H_2O_2$ source. Blankenship and Hartman[1] further developed a hypothesis that involves $H_2O_2$ as a transitional electron donor in a two-electron reaction releasing oxygen within a primitive system composed of an anoxygenic photosynthetic reaction center and a binuclear Mn-catalase enzyme (R2).

$$2H_2O_2 \rightarrow 2H_2O + O_2 \qquad (2)$$

Regardless of the disputes on the structural homology between the binuclear Mn-catalase and four Mn-containing core of the WOC[2,10], electrons in $H_2O_2$ can also be extracted by the Mn cluster that might directly evolve from inorganic Mn complexes[11], such as manganese bicarbonate[12] and a variety of $MnO_2$ minerals[13]. Thus, $H_2O_2$ could have played a key role in the developmental stage of oxygenic photosynthesis, but for that process to endure, it would have depended on the continuous availability of an exogenous source of $H_2O_2$.

Several abiotic geochemical sources of $H_2O_2$ were proposed to have existed on the early Earth[14–18]. For instance, Kasting[14,19] proposed a photochemical model in which $H_2O_2$ was produced from the photolysis of $H_2O$ in the early stratosphere and reached the ground via precipitation. The flux of $H_2O_2$ on the surface of the early Earth was calculated to be ~$10^6$ molecules $cm^{-2} s^{-1}$ based on a 0.2-bar $CO_2$ photochemical model[20], resulting in a dissolved $O_2$ concentration maximum of 0.08 nM in the surface water when all $H_2O_2$ decomposed to $O_2$ and $H_2O$. Pecoits et al.[21] similarly calculated a rate of ~$10^6$ molecules $cm^{-2} s^{-1}$ assuming lower $pCO_2$ of 0.01–0.1 bar. These theoretical calculations[14,19–21] collectively suggest that the ground level of photochemically produced $H_2O_2$ is very low because of the low atmospheric $H_2O$ vapor and the short atmospheric lifetime of $H_2O_2$ against ultraviolet photolysis in the stratosphere. Importantly, the trace levels of $H_2O_2$ by photochemical process alone would be insufficient to fuel the respiration of the smallest cells (3 nM dissolved $O_2$), yet high enough for developing a defense against oxygen toxicity[20,22]. Although it has been proposed that substantial photochemically produced $H_2O_2$ stored in glaciers might have been released to the oceans following a Snowball Earth event[23], the occurrence of possibly the earliest extensive glaciation (the Pongola glaciations at ~2.9 Gyr)[24] is still much later than the development time of oxygen-utilizing enzymes (~3.1 Gyr)[25].

Here we report experimental evidence for an overlooked mechanism for producing a stable abiotic source of reactive oxygen species (ROS, including $H_2O_2$ and •OH) involving water reacting with newly abraded surfaces of quartz ($SiO_2$). Specifically, $O_2$ and ROS are produced by reactions between water and the surface-bound radicals (SBRs), such as $\equiv SiO•$ and $\equiv SiOO•$, under strictly anoxic conditions. These SBRs are highly reactive surface defects on quartz and various other types of silicate minerals (e.g., pyroxene)[26,27] that can be created by the mechanical homolysis of Si–O bonds[28] through hydrological processes that induce quartz abrasion (e.g., currents, waves, and tides). We hypothesize that the persistent generation of $H_2O_2$ at quartz-water interfaces in rivers, estuaries, and deltas could have provided a source of $H_2O_2$ that stimulated the emergence of oxygenic photosynthesis from anaerobic predecessors on the early Earth[1,6].

## Results and discussion

**The SBR-induced generation of ROS at the abraded quartz-water interface.** We selected quartz for examination because it is the most abundant weathering end-product mineral in terrestrial environments and there is >3.2 Ga record of extensive quartz sandstones in high-energy aqueous environments[29]. Its basic unit, the $[SiO_4]^{4-}$ tetrahedron, is the structural unit of all silicate minerals. Si–O bonds in $[SiO_4]^{4-}$ tetrahedrons are mixtures of both ionic and covalent bonds[30], thus the rupture of Si–O bonds can produce SBRs with unpaired electrons (such as $\equiv SiO•$ and $\equiv SiOO•$)[31] which can split water molecules and produce ROS. Previous studies investigated the formation of radicals on the surface of pulverized minerals and their role in pathogenicity[32–37], which suggest that SBRs and the ROS produced are sensitive to the presence of atmospheric $O_2$ and $H_2O$. To avoid the false positive results induced by pre-existing $O_2$, we performed ball-milling of quartz sands (0.25–0.6 mm) in an $O_2$-free, $N_2$ atmosphere ($O_2 < 0.1$ ppm) and obtained a fine powder of quartz with a median particle size of 0.002 mm (Supplementary Fig. 1).

We investigated the feasibility of the formation of SBRs on the surface of the abraded quartz on Earth before its atmosphere accumulated free oxygen, e.g., the Great Oxidation Event (GOE), which occurred at ~2.45 to 2.3 Gyr ago[38,39]. We observed that the abrasion of quartz led to simultaneous formation of charged surface sites through heterolysis (i.e., $\equiv Si–O^-$ and $\equiv Si^+$, as per R3) and SBRs with unpaired electrons through homolysis (i.e., $\equiv Si–O•$, and $\equiv Si•$, and $\equiv Si–O–O•$, as per R4–5)[28,40] on the quartz surfaces (Fig. 1a). In turn, this led to the generation of E′ center ($\equiv Si•$), peroxy radical ($\equiv SiOO•$), and superoxide ion ($\equiv Si^+O_2^{•-}$) species. By contrast, there is no SBR on the aged surface of intact quartz. The SBRs were derived from the homolysis of $\equiv Si–O–Si\equiv$ (intrinsic bonds in quartz) and $\equiv Si–O–O–Si\equiv$ (a peroxy linkage formed during mechanical deformations)[41].

$$\equiv Si-O-Si\equiv \; \rightarrow \; \equiv Si-O^- + \equiv Si^+ \qquad (3)$$

$$\equiv Si-O-Si\equiv \; \rightarrow \; \equiv Si-O• + \equiv Si• \qquad (4)$$

$$\equiv Si-O-O-Si\equiv \; \rightarrow \; \equiv Si-O-O• + \equiv Si• \qquad (5)$$

X-ray diffraction (XRD) analyses were performed to further examine whether the generation of SBRs were related to structural changes during quartz abrasion. The XRD patterns show that the abraded quartz has distinctive variations in its quintuple lines that are consistent with the presence of crystal

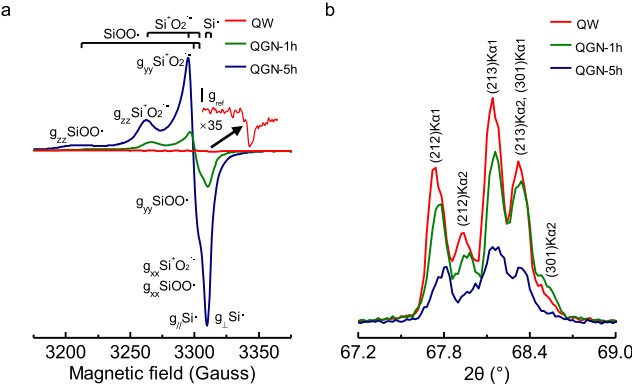

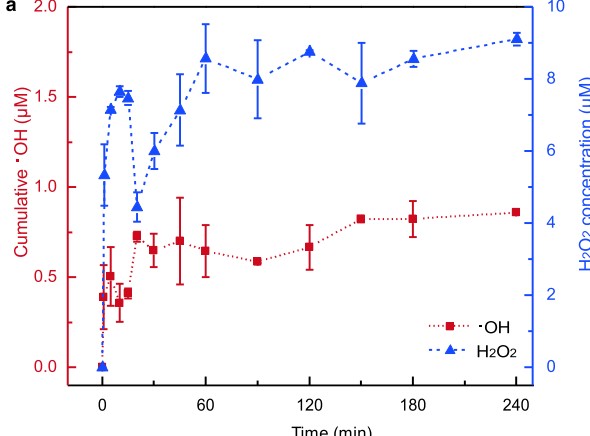

**Fig. 1 Mechanically induced surface-bound radicals and structural amorphization of quartz. a** EPR spectra of quartz before and after being ground in $N_2$ for 1 and 5 h. Three types of surface-bound radicals were identified, i.e., peroxy radical ($\equiv SiOO\bullet$, $g_{zz} = 2.064$; $g_{yy} = 2.007$; $g_{xx} = 2.002$), surface-bound superoxide ion ($\equiv Si^+O_2^{\bullet-}$, $g_{zz} = 2.044$; $g_{yy} = 2.010$; $g_{xx} = 2.002$), and E′ center ($\equiv Si\bullet$, $g_{//} = 2.001$; $g_\perp = 2.000$). **b** XRD patterns of quartz washed by 10 wt.% hydrofluoric acid (QW) and quartz ground in $N_2$ atmosphere for 1 h (QGN-1h) and 5 h (QGN-5h).

defects (Fig. 1b), i.e., the five characteristic diffraction peaks of quartz crystals changed from sharp-and-narrow to weak-and-broad of those with the abrasion caused basal slip[41]. Correspondingly, transmission electron microscopy (TEM) observations demonstrated that the amorphous layer that formed on the surface of the abraded quartz increased with a longer time of ball-milling (Supplementary Fig. 2). More importantly, electron paramagnetic resonance (EPR) measurements (Fig. 1a) showed that the increase in E′ center ($\equiv Si\bullet$) and oxygen radicals ($\equiv SiO\bullet$ and $\equiv SiOO\bullet$) was related to the increment in the thickness of the mechanically induced amorphous layer on the surface of the crystalline quartz.

We then examined the interaction between the resultant SBRs and water by submerging the abraded quartz into the oxygen-free water in a glovebox under an ultra-pure nitrogen atmosphere. We observed that the reaction led to the generation not only of ROS, such as hydroxyl radicals ($\bullet OH$) and $H_2O_2$ (Fig. 2a), but also of dissolved oxygen (Fig. 2b). The accumulation of these oxidants was demonstrated by a concomitant increase in the redox potential (from ~0 to 150 mV) at a near-constant pH (pH ≈ 3.77; Fig. 2b).

Measurements of the kinetics of ROS and $O_2$ release under anoxic conditions at the abraded quartz-water interfaces showed three stages. During the first stage (0–5 min), the reaction between SBRs and $H_2O$ led to a rapid increase in the concentrations of $\bullet OH$, $H_2O_2$, and dissolved $O_2$ in the suspension, resulting from reactions R6–R10.

$$\equiv SiO\bullet + H_2O \rightarrow \equiv SiOH + \bullet OH \qquad (6)$$

$$2\cdot OH \rightarrow H_2O_2 \qquad (7)$$

$$\equiv SiOO\bullet + H_2O \rightarrow \equiv SiOH + HO_2\bullet \qquad (8)$$

$$\equiv Si^+O_2^- + H_2O \rightarrow \equiv SiOH + HO_2\bullet \qquad (9)$$

$$HO_2\bullet \rightarrow 0.5H_2O_2 + 0.5O_2 \qquad (10)$$

During the second stage (5–120 min), fluctuations in the concentrations of ROS and dissolved $O_2$ were observed (Fig. 2b). The first slowdown in the increase of cumulative $\bullet OH$ concentration was attributed to a decrease in the exposure rate of SBRs, resulting in the suppression of R6. In turn, the decreased

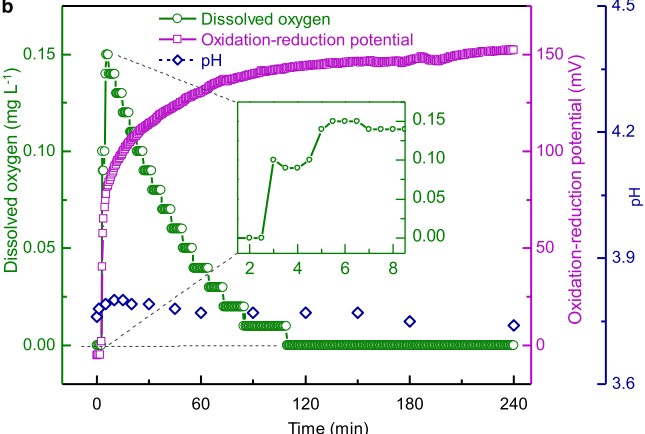

**Fig. 2 Time-course of $\bullet OH$, $H_2O_2$, and $O_2$ concentrations in quartz-water interface reactions. a** The cumulative $\bullet OH$ (filled square) and the concentration of $H_2O_2$ (filled triangle) in the suspension of quartz ground in $N_2$ atmosphere for 5 h. The error bars present standard deviations of two independent replicates. **b** The variations of dissolved oxygen concentrations (open circle), oxidation-reduction potential (open square), and pH (open diamond) in the reactions. All the independent experiments were conducted in duplicate using water with dissolved oxygen <0.01 mg L$^{-1}$ in a glovebox with $N_2$.

[$\bullet OH$] restricted the generation of $H_2O_2$ (R7). Moreover, both the decrease of SBR exposure (R8–R9) and consumption of $H_2O_2$ by other radicals (e.g., $\equiv Si^+O_2^{\bullet-}$) via R11 – the Haber-Weiss reaction[28] – resulted in a decreased [$H_2O_2$]. The subsequent increase in [$H_2O_2$] was negatively correlated to the concentration of the dissolved $O_2$. This is due to the dissolved $O_2$ produced in the intermediate reactions (R10–R11) reacted with $H\bullet$ to form $HO_2\bullet$ (R12)[42], in which the $H\bullet$ formed via the reaction between $H_2O$ and $\equiv Si\bullet$ (R13)[43]. The continuously generated $HO_2\bullet$ then gradually transformed to $H_2O_2$ in half-quantity cycling between R10 and R12, in which the dissolved $O_2$ was reduced to half of its initial amount in each cycle, concomitant with the generation of an equivalent molar amount of $H_2O_2$. Meanwhile, as predicted by Henry's law, ~15% dissolved $O_2$ might escape from the quartz-water interface to the headspace of the oxygen-free atmosphere (See Supplementary Text).

$$\equiv Si^+O_2^- + H_2O_2 \rightarrow \equiv SiOH + \bullet OH + O_2 \qquad (11)$$

$$H\bullet + O_2 \rightarrow HO_2\bullet \qquad (12)$$

$$\equiv Si\bullet + H_2O \rightarrow \equiv SiOH + H\bullet \qquad (13)$$

During the third stage (120–240 min), the concentrations of $H_2O_2$ approached steady values, and the concentration of the dissolved $O_2$ decreased to <0.01 mg L$^{-1}$. This indicates that most SBRs were consumed in the quartz-water interface reactions. The production of both $\bullet$OH and $H_2O_2$ was restricted by the quantity of SBRs as the concentration of the ROS was linearly correlated to the particle loading (Supplementary Fig. 3). About 47.13% of $H_2O_2$ was derived from reactions between $\equiv SiOO\bullet$ and water, while the remainder of the $H_2O_2$ was formed via the recombination of $\bullet$OH (Supplementary Fig. 4). In addition, the surplus H$\bullet$ combined with each other to form $H_2$ and escaped to the headspace (Supplementary Fig. 5).

Further, we confirmed that the intermediate $O_2$ in the reaction system could convert to ROS via the following two conditions. When quartz was ground in the presence of $O_2$, both the yielding of $\bullet$OH and $H_2O_2$ in the suspensions under various pH values were greater than those of quartz ground in an $O_2$-free atmosphere (Supplementary Fig. 6a). The increased ROS production was attributed to a higher proportion of oxygenic SBRs that were formed via the interaction between the initial SBRs and atmospheric $O_2$ as per R14[44]. On the other hand, an increase in the pre-existing dissolved $O_2$ concentration in water could enhance the production of $H_2O_2$ while having little effect on that of $\bullet$OH (Supplementary Fig. 6b), as depicted in R12 and R10.

$$\equiv Si\bullet + O_2 \rightarrow \equiv SiOO\bullet \qquad (14)$$

**Continuous generation of ROS in hydrodynamic environments with the abrasions of quartz.** Although our experiments show that 1 m$^2$ of abraded quartz surfaces could yield 9.65 nmol of $H_2O_2$ at a rate of ca. $1.95 \times 10^{11}$ molecules cm$^{-2}$ s$^{-1}$ (Fig. 2a and Supplementary Figs. 3 and 7), the true ROS release rate in nature is dependent on the intensity and extent of abrasion. We experimentally simulated the dynamic abrasion of quartz to mimic the conditions that quartz grains would undergo in coastal zones subject to waves and tides (Supplementary Fig. 8 and Video). Being ground mildly in a tumbling barrel, the particle size of quartz gradually decreased and the initially clear water became cloudy. The newly produced quartz surfaces kept increasing in a near-linear manner, which indicated that 129.75 m$^2$ of surface area could be produced by abrading 1 g of quartz for 1 year. Based on the laboratory abrasion rate of quartz and the subsequent $H_2O_2$ yield per unit area, the simulated $H_2O_2$ production rate is calculated to be $2.39 \times 10^{10}$ molecules g$^{-1}$ s$^{-1}$.

The EPR spectra displayed signals of spin adducts of radicals, including $\bullet$OH, $\bullet O_2^-$ (superoxide radical) and $^1O_2$ (singlet oxygen) during the time-course of quartz abrasion (Fig. 3). Additionally, the intensity of these signals increased along with the abrading time. These observations further support that ROS and $O_2$ could be continuously generated in aqueous environments with strong hydrodynamic processes.

**The $H_2O_2$ flux and the resultant redox evolution in the Archean.** Our results reveal a geochemical pathway for the generation of ROS and free oxygen at the quartz-water interface in aquatic environments that has not been previously described. As these reactions occurred naturally in the geological past, it stands to reason that a stable and persistent source for ROS and $O_2$ was generated in subaqueous conditions in the absence of biological processes, such as oxygenic photosynthesis. Crucially, with a higher yield and better accessibility, this oxygen-producing

pathway not only provided a favorable source of $H_2O_2$, but the $O_2$ ultimately produced would have exerted a new evolutionary pressure to early anoxygenic photosynthesizers[6].

We extrapolated our experimental results to depositional scale and calculated the $H_2O_2$ flux in the turbulent subaqueous environments. Our calculations show that an Archean river carrying a heavy load of suspended sediments could potentially transport $H_2O_2$ to the oceans at a flux of $5.81 \times 10^{14}$ molecules cm$^{-2}$ s$^{-1}$, while the in-situ flux of $H_2O_2$ at the Archean delta/shore could additionally reach to $4.87 \times 10^{11}$ molecules cm$^{-2}$ s$^{-1}$ (See Supplementary Text and Fig. 9). To further put these numbers into perspective, we compared our results with two endmember $O_2$-producing reactions: atmospheric photochemical processes (minimum value) and benthic cyanobacterial mats (maximum value). In the case of the former, our calculations suggest that in such coastal environments, atmospheric photochemical processes could yield $5 \times 10^5$ molecules $O_2$ cm$^{-2}$ s$^{-1}$) (based on R2). By contrast, a literature survey of 84 in-situ measurements of oxygenic photosynthesis in benthic microbial ecosystems showed that depth-integrated net rates ($O_2$ produced – $O_2$ immediately consumed) are log-normally distributed and generally fall within two orders of magnitude ($10^0$–$10^{-2}$ nmol cm$^{-2}$ s$^{-1}$) regardless of the benthic environment[45]. With this range in mind, those authors adapted the median net $O_2$ production rate of 0.16 nmol cm$^{-2}$ s$^{-1}$ or $9.63 \times 10^{13}$ molecules cm$^{-2}$ s$^{-1}$. This rate of $O_2$ production from modern cyanobacterial mats is ~400 times higher than the in-situ flux of $O_2$ at the Archean delta/shore ($2.44 \times 10^{11}$ molecules cm$^{-2}$ s$^{-1}$; based on R2 and using the in-situ $H_2O_2$ production rate of $4.87 \times 10^{11}$ molecules cm$^{-2}$ s$^{-1}$). However, it clearly demonstrates that the abrasion of quartz sands in hydrodynamic processes can act as a geologically significant oxygen-producing pathway.

Such high yielding of oxidants could have created locally oxidized environments that not only were detrimental to the early anaerobic photoautotrophs growing within the microbial mats, but the $O_2$ would eventually have expanded into the water column and readily eliminated the reductants necessary for anaerobic photosynthesis by planktonic communities (e.g., $H_2$, $HS^-$, $Fe^{2+}$). We modeled how this $O_2$ first effected the photosynthetic communities growing within a mat, and then extrapolated to the water column above the continental shelf. In terms of the microbial mats, $O_2$ generation would have presented a challenge to microbial metabolism that used electron donors that are readily oxidized. Most cyanobacteria are sensitive to sulfide as it directly inhibits the water-splitting reaction of Photosystem II (PSII)[46]. Even in metabolically versatile cyanobacteria mats, with a $H_2S$ concentration as low as 1 μM, the oxygenic photosynthesis is severely inhibited and replaced by sulfide-oxidizing anoxygenic photosynthesis[47,48]. With the depletion of sulfide by diffusing $O_2$, the onset of oxygenic photosynthesis can occur, then the competition with the two metabolisms (anoxygenic photosynthesis and chemosynthesis) is possible.

In the case of the water column, and taking water-soluble Fe(II) as an example, it has been proposed that the oldest banded iron formations (BIF) were formed via the activity of anoxygenic photosynthetic bacteria, the photoferrotrophs[49,50]. These bacteria might have grown throughout the photic zone prior to cyanobacterial evolution, but thereafter were progressively marginalized to greater depths as the photic zone became more oxygenated[51]. Using a quantitative model modified from McKay and Hartman[6], we performed the time evolution of $H_2O_2$ concentration in the oxic zone and the length of the pathway to which oxic conditions extend into the Archean shallow seawater near deltas and shores based on the two $H_2O_2$ flux calculated above (Supplementary Figs. 9–10). The continuing source of ROS and $O_2$ can produce an oxidized front moving

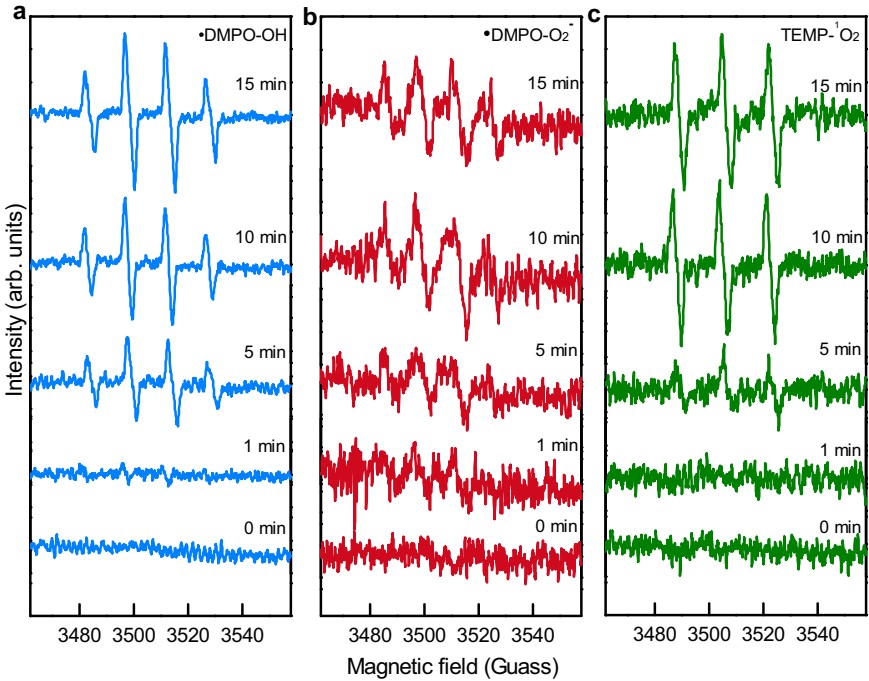

**Fig. 3 Continuous generation of •OH, •O₂⁻, and ¹O₂ during dynamic fracturing processes of quartz in aquatic environments.** The characterized EPR signals of •OH, •O₂⁻, and ¹O₂ spin adducts occur and grow with increased ball-milling time (0–15 min). **a** Hydroxyl radical (•OH) spin adducts (•DMPO-OH), a 4-fold characteristic peak with an intensity ratio of 1:2:2:1. **b** Superoxide radical (•O₂⁻) spin adducts (•DMPO-O₂⁻), a 4-fold characteristic peak with an intensity ratio of 1:1:1:1. **c** Singlet oxygen (¹O₂) spin adducts (•TEMP-¹O₂), three peaks with an intensity ratio of 1:1:1.

outward from the delta to the open ocean that was characterized by ferruginous seawater[52].

**The H₂O₂-driving evolution of photosynthesis.** The O₂ levels within the oxic zone are several orders higher than the threshold stimulating the development of oxygen tolerance (e.g., superoxide dismutase, Mn catalase) and aerobic respiration (dissolved O₂ > 3 nM) in the anoxygenic phototrophs[6,22,53]. Furthermore, it provides the indispensable oxidizing species (i.e., O₂ and H₂O₂) for the key evolution of oxygenic photosynthesis. Due to the catalysation of aerobic cyclase enzyme in the organisms, the mechanical/chemically produced O₂ could have served as the earliest external substrate for the O₂-dependent biosynthesis of chlorophyll-*a* precursor Mg-divinyl chlorophyllide[54,55]. In summary, the primitive PSII composed of an intermediate pigment of chlorophyll-*a* and an original Mn₂-cluster (e.g., the Mn catalase or MnO₂ minerals) would have allowed the ancestral cyanobacteria to extract two electrons from H₂O₂ (the transitional electron donors) at a time, liberating the earliest biogenic O₂ (R2)[1,11,55,56].

Phylogenetic analyses of the early evolution of the ROS gene network proteins suggest that superoxide dismutase, catalase, and peroxiredoxins evolved in the presence of ROS ~4.1–3.6 Gyr ago[57], well before the geochemical evidence of the rise of oxygen-evolving photosynthetic cyanobacteria[58–60]. These ROS-scavenging enzymes might have emerged in the initial adaptation to the weakly oxic environments and crucially protected the earliest cyanobacteria from the toxicity of ROS[61]. Notably, the rise in both oxygenases and other oxidoreductases suggests a wider availability of trace oxygen ~3.1 Gyr[25]. Following on from this, we propose that the early evolution of oxygenic photosynthesis might have originated as benthic microbial communities at river or coastal setting[45,56]. In such high-energy hydrodynamic environments, the continuous abrasion of quartz-bearing sediment provided a high yield of oxidants despite

the contemporary atmosphere being anoxic (O₂ < 10⁻⁷ atm[62]). This process would have operated independent of atmospheric conditions, similar to the benthic pre-GOE oxygen oases previously proposed[45]. Moreover, in the absence of plant root systems, Archean surface terrains would have been subject to rapid migration of riverbeds and a predisposition for wide, braided streams to transport and disperse the large volumes of supermature quartz-rich sands weathered from the source terrains[63]. Subjected to much more intense tidal erosion than today[64], the Archean land surface would have experienced enhanced wetting, conditions favouring colonization by microbial mats. With the mechanically resistant and sandy biolaminae shielding the harmful ultraviolet light, the ancestral cyanobacteria could perform photosynthesis via the H₂O₂ delivered by currents waves and tides associated with these high-energy environments (Fig. 4a).

It is perhaps unsurprising then that some of the best evidence for Archean life comes from the 3.22 Gyr Moodies Group in South Africa, where extensive microbial mats draped fluvial conglomerates, gravelly sandstones, and shallow marine siliciclastic tidal deposits[65,66]. Wavy and crinkly carbonaceous laminations within these gravelly sandstones, and on top of the conglomerate beds, are often bent upwards and plastically deformed by subvertical fluid-escape structures which indicate their cohesive water-impermeable nature and synsedimentary origin. Petrographic analysis further reveals that during periods of increased current velocity, laminae were partially eroded, ripped up, and reworked as fragments up to several centimetres in length. Although not conclusive of cyanobacteria, δ¹³C_org values ranging between −23.6‰ and −17.9‰ are consistent with the Calvin-Benson cycle, while δ¹⁵N values of the terrestrial mats with a mean of +4.3‰ are suggestive of partial nitrification and/or denitrification[66], both of which require oxidative nitrogen cycling in the presence of free oxygen in the mats[66,67]. Moreover, the contemporaneous S-isotopic compositions in the Moodies

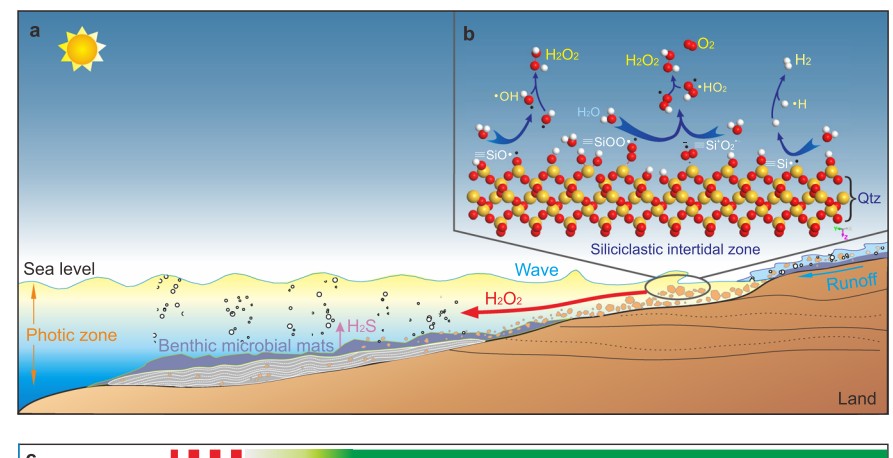

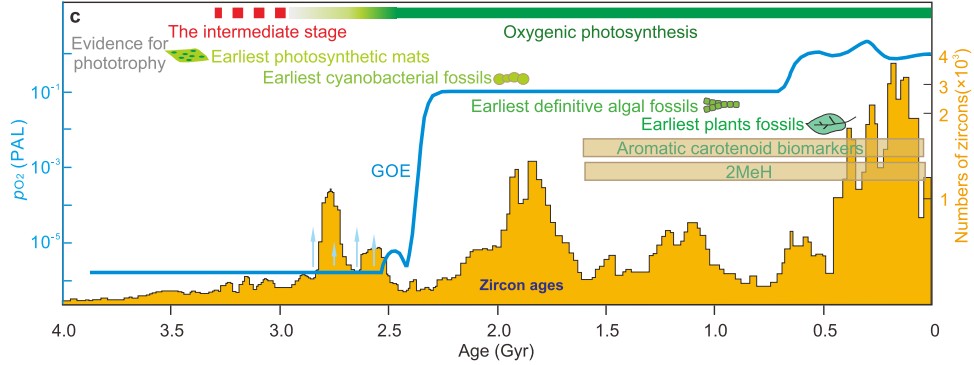

**Fig. 4 A plausible origin of oxygenic photosynthesis from an anaerobic photosynthetic community with the supply of H₂O₂ released at mineral-water interface along the siliciclastic intertidal zone. a** The H₂O₂-induced evolutionary impetus on the organisms that conduct anoxygenic photosynthesis in microbial mats. **b** H₂O₂ generated at the quartz-water interface in the high-energy hydrodynamic intertidal zone. **c** Geological constraints on the origin of oxygenic photosynthesis including data from the body and molecular fossil (lipid biomarker) records, the age distribution of zircons and the oxygen concentration relative to the present atmospheric level (PAL). Modified from Fischer[4], Lyons[58], and Hawkesworth[69].

Group barites also suggest some pyrite weathering under weakly oxidizing conditions[68].

Our hypothesis neatly ties together a significant source of abiotic oxygen on the Archean Earth and the subsequent evolution of oxygenic photosynthesis. As more quartz-rich rocks (such as tonalite-trondhjemite-granodiorite) emerged in the continental landmass of the Archean[69] (Fig. 4b), the quartz-rich rocks suffered aggressive erosion during fluvial transport to the coast. The increased oxidation (via H₂O₂ and O₂) placed an increasing evolutionary pressure on the anaerobic microbial world. Eventually, as the ancestral cyanobacteria developed oxygenic photosynthesis and gained the ability for extracting electrons from H₂O[1,56,70], they would have outcompeted the anaerobes that were constrained by a limited geological supply of reductant electron donors[71]. This shift in photosynthesis ultimately changed the structure of the Archean Earth's biosphere and initiated oxidative weathering on land[45,72]. Our study does not constrain the time when oxygenic photosynthesis successfully evolved but it does provide a plausible scenario for the initial development of oxygenation on the early Earth. Significantly, from a geochemical perspective, this means that some existing Archean proxies of O₂ are not necessarily tied to early oxygenic photosynthesis.

## Methods

**Sample preparation and storage.** Quartz sands were purchased from Richjoint (Shanghai, China; SiO₂ > 99 wt.%). The quartz raw material was mesh-sieved and followed by a hydrofluoric acid (10 wt.%) washing for 8 h to remove surface impurities and produce the surface without dangling bonds (QW)[28]. The 0.25–0.6 mm quartz particles were chosen and washed with distilled deionized water for a few times until the pH of the cleaning solution reached neutral, then

dried in oven at 110 °C for 24 h and stored in a glovebox for further experiments (N₂ > 99.999%, H₂O < 0.1 ppm, O₂ < 0.1 ppm, Mikrouna).

In the glovebox, the quartz samples were loaded into a zirconia ceramic jar with two inlet switches, and sealed tightly with a gasket ring and screw fasteners. Then, the jar was filled with the same atmosphere as in the glovebox before being transferred to a planetary ball mill (Fritsch, Pulverisette 6) for grinding at 350 rpm for 1 and 5 h, respectively. After grinding, the jar was moved back to the glovebox, and the abraded quartz samples (QGN-1h and QGN-5h) were collected and sealed in glass bottles in the glovebox to protect the surface reactive sites.

A control sample was ground in a pure O₂ atmosphere for 5 h with the following procedure: first, quartz powders were loaded into the jar in the glovebox and the jar was sealed tightly with the aforementioned procedure. Then, the jar was moved out of the glovebox and vacuumed through one inlet with a mechanical pump, and the jar was filled with pure oxygen (>99.99%) for grinding by the aforementioned procedure. Lastly, the sample was transferred to glass bottles in a glass desiccator full-filled with ultrapure N₂. This sample was used in the according experiments within 48 h after grinding.

**Experimental procedures.** All waters used in the experiments were deionized to 18.2 MΩ cm⁻¹ at 25 °C. The waters with a range of dissolved oxygen contents (DO = 0.01, 0.64, 1.28, and 4.10 mg L⁻¹, respectively) were prepared by diluting the high DO (9.00 mg L⁻¹) water with low DO (<0.01 mg L⁻¹) water. Potassium phosphate buffer solutions of various pH values (pH = 4.0, 6.0, 7.0, and 7.4) were prepared by mixing 1/15 M KH₂PO₄ with 1/15 M K₂HPO₄ solutions with according ratios.

Measurements of the reaction kinetics of ROS generation were conducted in the glovebox at room temperature. A 250-mL solution (DO < 0.01 mg L⁻¹) with 10 mM benzoic acid (BA, Aladdin, the trapping agent for hydroxyl radical (•OH)) was prepared and transferred to a three-neck flask with continuous magnetic stirring. Probes were then put into the solution through the flask necks to measure the pH values (Mettler-Toledo FiveEasy PlusTM), DO, and redox potentiality (HQ40, HACH), respectively. When 20.00 g of QGN-5h (the specific surface area is 5.21 m² g⁻¹) was quickly plunged into the solution, the reaction started and lasted for 4 h. In all, 4.00 mL of suspension was extracted and filtered through a 0.22-μm membrane (Jinteng) at each sampling time (0, 1, 5, 10, 15, 20, 30, 45, 60, 90, 120, 150, 180, and 240 min, respectively). These subsamples were used for measuring the concentrations of hydroxyl radical and hydrogen peroxide.

To examine the effects of atmosphere and aquatic chemistry on the production of ROS, suspensions were prepared by submerging the abraded quartz sample (1.00 g) into the deionized water (5.00 mL) with magnetic stirring. The QGN-5h sample was added into the water with various concentrations of dissolved $O_2$. The quartz samples ground under different headspace gases ($N_2$ or $O_2$) were then mixed with 5.00 mL of potassium phosphate buffer solutions that were previously prepared under various pH conditions (DO < 0.01 mg $L^{-1}$). The filtered liquid samples were collected for measuring the concentrations of hydroxyl radical and hydrogen peroxide.

To further examine the effect of specific surface area on the production of ROS, the loading of quartz (i.e., the ratio of solid mass to water volume (DO < 0.01 mg $L^{-1}$)) was set at 0.10, 0.20, 0.40, 0.60, 0.80, and 1.00 g $mL^{-1}$, respectively. All filtered liquid samples from the batch experiments were collected for ROS measurements.

To confirm the presence of $\equiv SiOO\bullet$ on the surface of the abraded quartz produced in anoxic conditions, we conducted control experiments by scavenging specific radicals. Methanol was chosen as the $\bullet OH$ scavenger, while benzoquinone was used to scavenge $\bullet O_2^-/HO_2\bullet$ (Burns et al.[73]). In total, 1.00 g of the abraded quartz sample (QGN-5h) was plunged into 5.00 mL of the solution (DO < 0.01 mg $L^{-1}$) with 2 mM the ROS scavenger (methanol or benzoquinone) with magnetic stirring. All filtered liquid samples were collected to measure the concentrations of $H_2O_2$.

To experimentally test if ROS could be continuously produced in the natural erosion of quartz by waves and tides, a tumbling barrel was used to grind quartz (Supplementary Fig. 8a–b). Such an apparatus has been used to simulate the fluvial abrasion of small pebbles and sands in laboratory[74,75]. The solid concentration (quartz sands with a size of ca. 2 mm) is set to 250 kg $m^{-3}$, and the tumbling barrel was rolling at 1 m $s^{-1}$. About 10 mL of the suspension was sampled at 24, 48, 72, 96, and 120 h, respectively. After centrifugation and drying, the quartz powders were collected for surface area measurements.

EPR spin trapping techniques were used to examine the transient radicals generated during the dynamic fracturing process of quartz. Probes with specific spin were selected to react with transient free radicals[73]. Hydroxyl radical ($\bullet OH$), superoxide radical ($\bullet O_2^-$), and singlet oxygen ($^1O_2$) were trapped by DMPO (5,5-dimethyl-1-pyrroline N-oxide), DMPO/DMSO (dimethyl sulfoxide), and TEMP (2,2,6,6-tetramethylpiperidine), respectively, to form more stable spin adducts for quantification. In all, 1.00 g of quartz sand was ground in 2.00 mL of solutions containing the trapping agent (10 mM) by vibrating ball mill (Pulverisette 23 Mini-Mill). After shaking at 50 Hz frequency for certain times (0, 1, 5, 10, and 15 min), 0.50 mL of the suspension containing spin adducts was sampled and filtered for EPR measurements.

The headspace gas composition in the jar during the quartz grinding process was monitored. In total, 30.00 g of quartz sands and 10.00 mL water (DO < 0.01 mg $L^{-1}$) were loaded into the jar in the glovebox. The sealed jar filled with the same atmosphere as in the glovebox (i.e., ultrapure $N_2$) was taken out, and transferred to a planetary ball mill (Fritsch, Pulverisette 6) for grinding at 350 rpm for certain times (0, 0.5, 1, 1.5, 2, 2.5, 3, and 5.5 h). In total, 1.50 mL gas in the headspace of the jar was extracted with a syringe from the valve with a thick silicone pad for gas content measurement in each interval.

**Characterization of the crystallography of quartz.** XRD patterns were obtained over the 2θ range from 3 to 70° on a Bruker D8 ADVANCE X-ray diffractometer, operated at 40 kV and 40 mA with Cu Kα radiation. The scanning step was 3° $min^{-1}$ (2θ).

Surface area measurements were performed with the BET $N_2$ technique. Nitrogen adsorption-desorption measurements were conducted at 77 K using an ASAP 2020 Surface Area & Pore Size Analyzer (Micromeritics Instrument Corporation). Prior to the measurement, samples were degassed in vacuum at 200 °C for 12 h.

**Characterization of the particle size and morphology of quartz.** The particle size of the quartz powder (QGN-5h) was measured with a laser particle analyser (Chengdujingxin, JL-1177). The morphology of the quartz powder (QGN-5h) was observed with a scanning electron microscope (SEM, Phenom XL).

**Measurements of surface-bound radicals.** Surface-bound radicals formed via grinding were measured by EPR on a Bruker A300-10-12 spectrometer. The settings for the EPR measurements were as follows: center field, 3320G; sweep width, 500G; microwave frequency, 9.297 GHz; modulation frequency, 100 kHz; power, 1.0 mW; and temperature, 77 K.

**Measurements of headspace gas products.** The gas products in the headspace of the sealed jar – after the quartz sands were ground in water – was determined by gas chromatography (GC, Agilent Technologies 7890-0322). Each gas sample was injected into the GC inlet connected with a vacuum glass system. The gas analyser included two thermal conductivity detectors for the analysis of permanent gases and a flame ionization detector for the analysis of hydrocarbon gases, as well as five rotary valves and seven columns. This enabled the analysis of all gaseous components with a single injection.

**Measurements of hydroxyl radicals.** All experiments were carried out at 25 ± 2 °C in the dark. Quartz samples were mixed with the solutions containing the $\bullet OH$ trapping agent (benzoic acid, BA, 10 mM), then filtered and sampled for $\bullet OH$ measurements by determining the concentrations of the oxidative product (p-Hydroxybenzoic acid, p-HBA) of benzoic acid (BA)[76]. For the determination of p-HBA, 1.00 mL of a filtered liquid sample was rapidly mixed with 1.00 mL methanol (chromatographic grade, J&K Chemical) to quench further oxidation caused by $\bullet OH$. The p-HBA concentration was measured by a high-performance liquid chromatography (HPLC, Agilent 1260) equipped with a UV detector and an Inter Sustain C18 column (4.6 × 250 mm). The mobile phase was a mixture of 0.1% trifluoroacetic acid aqueous solution and acetonitrile (65:35, v:v) at a flow rate of 1.00 mL $min^{-1}$. The volume of sample injection was 20 μL, the column temperature was 30 °C, and the detection wavelength of UV was set at 255 nm. The p-HBA retention time was 3.2 ± 0.2 min. The detection limit of p-HBA was optimized to 0.01 μM, which corresponds to 0.059 μM $\bullet OH$.

**Measurements of hydrogen peroxide.** The $H_2O_2$ concentration was determined using a modified version of Leuco Crystal Violet (LCV, spectrographic grade, John Long) technique optimized for UV-Vis measurements[77]. LCV can be oxidized by $\bullet OH$ generated from $H_2O_2$ with the catalysis of the enzyme horseradish peroxidase (HRP, type III, >300 units $g^{-1}$, Aladdin). The oxidizing product of LCV is crystal violet cation ($CV^+$), which has a spectrophotometric absorbance maximum at 590 nm. The absorbance of $CV^+$ was measured on a Shunyuhengping UV2400 spectrometer.

Reagents (all stored at 4 °C and brought up to room temperature of 25 ± 2 °C before analysis) were added to 1.70 mL filtered liquid sample in the following order for a total volume of 2 mL: 200 μL 100 mM $KH_2PO_4$ (Aladdin) as pH buffer, 50 μL 41 mM LCV (dissolved with HCl), and 50 μL 0.5 mg $mL^{-1}$ HRP (containing 1.5 mM azide from Aladdin to prevent bacterial growth). After shaking to homogenize the samples, they were kept in dark at room temperature (25 ± 2 °C) for 30 min to stabilize the absorbance. Absorbance measurements were taken in 1 cm path-length cuvettes.

## Data availability

The authors declare that the main data supporting the findings of this study are available in the following Data Repository link: https://doi.org/10.17632/h7y9jdmpxp.1. Source data are provided with this paper.

## Code availability

The code in this study is available at https://doi.org/10.5281/zenodo.5515787.

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

## Acknowledgements

We thank J. William Schopf, Robert E. Blankenship, Noah J. Planavsky, Christopher P. McKay, and anonymous reviewers for their helpful discussions and suggestions. H.H. and J.Z. were supported by National Natural Science Foundation of China (Grant No. 41921003), National Science Fund for Distinguished Young Scholars (41825003), CAS Key Research Program of Frontier Sciences (Grant No. QYZDJ-SSW-DQC023), and the Strategic Priority Research Program (B) of Chinese Academy of Sciences (Grant No. XDB18000000). Y.Li was supported by the Strategic Priority Research Program of Chinese Academy of Sciences (Grant No. XDB41000000). This is contribution No. IS-3096 from GIGCAS.

## Author contributions

H.H. and J.Z. led the projects and designed the experiments; X.W. performed the experiments with assistance from Y.Lv and Y.Y.; H.H., J.Z., X.W., and H.X. analyzed the data; H.H., X.W., Y.Li, and K.K. wrote the manuscript with input from H.X. and J.Z.

## Competing interests

The authors declare no competing interests.
