## [Peer Review File · Nature Communications]

An abiotic source of Archean hydrogen peroxide and oxygen that pre-dates oxygenic photosynthesisREVIEWER COMMENTS

Reviewer 1:

This is an interesting paper that proposes a substantial source of H₂O₂ on the early earth arising from weathering of quartz. The proposal is backed up with experimental results and modeling studies so that it is argued that this previously overlooked process could have produced sufficient H₂O₂ to serve as a transitional electron donor for a precursor of oxygenic phototrophs.

I found the idea and the data to be convincing. I have only a few suggested revisions.

Some cyanobacteria can use sulfide as an electron donor. This should be cited in the discussion on p.11.

Padan, E. (1979) Facultative anoxygenic photosynthesis in cyanobacteria. *Annual Review of Plant Physiology* 30: 27–40.

Liu, D., Zhang, J., Lu, C., Xia, Y., Liu, H., Jiao, N. Xun, L. Liu, J. (2020) *Synechococcus* sp. strain PCC7002 uses sulfide:quinone oxidoreductase to detoxify exogenous sulfide and to convert endogenous sulfide to cellular sulfane sulfur. *mBio* 11: e03420-19.

Another source of substantial H₂O₂ was proposed by Liang et al., which should be cited.

Liang, Mao-Chang Hartman, Hyman; Kopp, Robert E.; et al. (2006) Production of hydrogen peroxide in the atmosphere of a Snowball Earth and the origin of oxygenic photosynthesis. *PNAS* 103: 18896-18899

Reviewer 2:

This manuscript reports the generation of reactive oxygen species (ROS, such as H₂O₂) during the abrasion of quartz mineral surfaces. The implication is that the produced H₂O₂ can act as an electron donor for photosynthesis during Archean. The authors characterized the production of ROS during ball milling of quartz in the absence or presence of oxygen, and calculated the potential flux of H₂O₂ generated by such process in natural environments. The calculations and related discussions on the potential natural processes involving H₂O₂ is interesting. However, the production of ROS by mineral surfaces is a well-known phenomenon and has been extensively studied in both geochemistry and biomedical fields. For example, in a 2005 review by Schoonen et al (below), the mechanisms and experimental/computational evidences of quartz surface produced ROS is reviewed in detail. Thus it is unclear how this study contributes to new knowledge.

Schoonen et al. (2005) Mineral-Induced Formation of Reactive Oxygen Species. *Reviews in Mineralogy and Geochemistry*. DOI: 10.2138/rmg.2006.64.7

The strong mechanical force induced by ball milling is also not well justified. This is the key point for the production of surface defects and ROS. The particle size and morphology of the quartz before/after ball milling were not well characterized. The relevance of such strong mechanical grinding (both intensity and time span) to natural environments was not justified. The calculations were mainly based on quartz surface area, but the H₂O₂ flux is heavily dependent on the intensity and extent of grinding.

Specific comments:

L25, L36: abstract should not contain references.

L67-69. This statement should be revised. The formation of Mn(III,IV) oxide is typically considered to involve the presence of oxygen. But here the authors are discussing an anoxic environment.

Also, the Mn in Mn-catalase and WOC are in the reduced state, so the source of Mn(II) might be from direct dissolution of Mn(II)-containing minerals instead of redox reactions between Mn oxides and H₂O₂.

L103-104. Reference needed.

L121-123. Such peak broadening is more likely to be related to the reduction of particle size and decreasing crystallinity, a very common phenomenon for nanoparticles. Did the authors characterize the particle morphology and particle size before and after ball milling?

The experiments with O₂ showed positive correlation between ROS production and O₂ concentration. The authors indicated that the anoxic grinding experiments were conducted in N₂. However, commercial ultrapure N₂ gas typically contains trace amount of O₂ (a few ppm), which might be inductive to the reactions. Did the authors measure the oxygen concentration of the gas?

The grinding experiments were conducted outside the glovebox with the jars filled with N₂. Did the authors ensure that the jars were well-sealed during grinding? Since the authors focused on the anoxic production of ROS during grinding, testing of oxygen level or potential leaking is critical.

The grinding experiments were conducted in a ball mill at 350 rpm. How does this translate to mechanical forces related to natural systems such as rives and tides? The ball milling also generated amorphous layers on quartz surface, which directly contributes to surface defects and production of ROS. Should we expect natural grinding processes by tides or waves to generate such intensity and kinetics of surface and structure transformation?

Reviewer 3:

Summary

The He et al., manuscript demonstrates laboratory-based abiotic generation of hydrogen peroxide (H₂O₂) due to quartz grain abrasion, which is proposed to be a potential electron donor and oxidant that exerted an evolutionary pressure for the emergence of oxygenic photosynthesis in the Archean eon. The 3.22 Ga Moodies group microbial mats are also proposed to have geochemical evidence that potentially supports the presence of hydrogen peroxide production via the quartz abrasion mechanism that would have promoted the eventual evolution of oxygenic photosynthesis.

This is a very interesting study and mechanism for H₂O₂ production prior to the evolution of oxygenic photosynthesis. However, I think there needs to be a stronger link between the observed laboratory mechanism and natural conditions. There are several issues that should be addressed to warrant publication in Nature Communications:

It is not clear if the ball-milling of quartz in the lab is a realistic representation of natural quartz abrasion caused in turbulent waterways or coastlines. How do the abrasive forces of the experiment compare to sandy riverbeds, deltas, or wave impact zones?

Turbulent events/conditions that are proposed to result in H₂O₂ production are often episodic. Any reduction in H₂O₂ production would result in rapid dilution of H₂O₂ to coastal waters. How would temporal variation impact H₂O₂ concentrations and impact on microbial communities?

The study indicates that H₂O₂ production by quartz abrasion is greater under oxic conditions. How does H₂O₂ production in the experiment compare with H₂O₂ levels in modern day oxic coastal waters, deltas, or turbulent rivers? How does H₂O₂ production in the lab compare with H₂O₂ production in the Yellow river in the quartz grain-mixing interface? Modern microbial communities of today are more extensive than in the Archean and would certainly impact H₂O₂ levels to a

greater extent than in the Archean, this could be discussed when comparing H₂O₂ production in modern day natural systems.

How would primitive tectonics taking place at 3.2 Ga impact H₂O₂ production? Could enhanced tectonic activity enhance H₂O₂ production?

Minor changes:

Abstract

Lines 22 to 24 is an awkward sentence, consider changing to something like: "Anoxygenic predecessors used a variety of reductants, such as H₂, H₂S or Fe(II), but the manner in which photosynthesis transitioned to water as the electron donor remains unclear."

Lines 24 to 26 could be clearer, consider changing to something like: "An intriguing hypothesis proposes that hydrogen peroxide (H₂O₂) once acted as the electron donor as an intermediate stage¹ prior to the evolution of oxygenic photosynthesis, but its abundance during the Archean would have been limited."

Line 27: insert the word "production" after "H₂O₂"

Lines 34 to 37: the final sentence of the abstract is too passive. There should be more evidence stated here in the abstract to connect these microbial mats to the use of H₂O₂ as an electron donor (redox evidence described in the discussion).

Main Text

The main text needs to be separated into subsections with subheadings, including a conclusions section at the end. This will provide more structure to the manuscript.

Lines 40 to 41: change to "...biological innovation that allowed water to be used as an electron source and dioxygen gas..."

Line 106: change to "...the Great Oxidation Event (GOE)..."

Line 107: "Farquhar et al., 2000" should be cited here as well, and the range of "2.45 Ga to 2.3 Ga" is more appropriate than one time point at 2.45 Ga.

Atmospheric Influence of Earth's Earliest Sulfur Cycle

James Farquhar*, Huiming Bao, Mark Thiemens

Science 04 Aug 2000:

Vol. 289, Issue 5480, pp. 756-758

DOI: 10.1126/science.289.5480.756

Line 187: why is "(REF.)" needed here in parentheses? Could you just add reference 31 at the end of the sentence?

Line 314: delete "the" before "early oxygenic photosynthesis."

AUTHOR'S RESPONSE TO REVIEWERS

Comments from Reviewer #1

Comment 1: This is an interesting paper that proposes a substantial source of H₂O₂ on the early earth arising from weathering of quartz. The proposal is backed up with experimental results and modeling studies so that it is argued that this previously overlooked process could have produced sufficient H₂O₂ to serve as a transitional electron donor for a precursor of oxygenic phototrophs. I found the idea and the data to be convincing. I have only a few suggested revisions.

Response: Thank you very much for the positive comments.

Comment 2: Some cyanobacteria can use sulfide as an electron donor. This should be cited in the discussion on p.11.

Padan, E. (1979) Facultative anoxygenic photosynthesis in cyanobacteria. Annual Review of Plant Physiology 30: 27–40.

Liu, D., Zhang, J., Lu, C., Xia, Y., Liu, H., Jiao, N. Xun, L. Liu, J. (2020) Synechococcus sp. strain PCC7002 uses sulfide: quinone oxidoreductase to detoxify exogenous sulfide and to convert endogenous sulfide to cellular sulfane sulfur. mBio 11: e03420-19.

Response: Thank you for the suggestion. The two references have now been cited in the revised manuscript. (Lines 258 and 261).

Comment 3: Another source of substantial H₂O₂ was proposed by Liang et al., which should be cited.

Liang, Mao-Chang Hartman, Hyman; Kopp, Robert E.; et al. (2006) Production of hydrogen peroxide in the atmosphere of a Snowball Earth and the origin of oxygenic photosynthesis. PNAS 103: 18896-18899.

Response: Again, thank you. This reference has now been cited in the text (Lines 89–93).

Comments from Reviewer #2

Comment 1: This manuscript reports the generation of reactive oxygen species (ROS, such as H₂O₂) during the abrasion of quartz mineral surfaces. The implication is that the produced H₂O₂ can act as an electron donor for photosynthesis during Archean. The authors characterized the production of ROS during ball milling of quartz in the absence or presence of oxygen, and calculated the potential flux of H₂O₂ generated by such process in natural environments. The calculations and related discussions on the

potential natural processes involving H₂O₂ is interesting. However, the production of ROS by mineral surfaces is a well-known phenomenon and has been extensively studied in both geochemistry and biomedical fields. For example, in a 2005 review by Schoonen et al (below), the mechanisms and experimental/computational evidences of quartz surface produced ROS is reviewed in detail. Thus, it is unclear how this study contributes to new knowledge.

Schoonen et al. (2006) Mineral-Induced Formation of Reactive Oxygen Species. *Reviews in Mineralogy and Geochemistry*. DOI: 10.2138/rmg.2006.64.7.

Response: Thank you for your comments. The reviewer is correct in that we should have referenced more work about the mineral surface reactivity. In our attempt to stay within the reference limits, we simply failed to be as thorough as we should have been. We feel that we have rectified this now by adding more references pertaining to mineral surface reactivity, such as Fubini et al. (1999, 2003), Cohn et al. (2006), Harrington et al. (2012), Xu et al. (2013) and Kaur and Schoonen (2017)¹⁻⁶ (Lines 113 to 116). Collectively, these references demonstrate the specific surface reactivity of pulverized minerals with water and how the ROS release at these surfaces and the role of ROS in the pneumoconiosis pathogenesis. However, what is unique in our experimental study is that we aimed at disclosing an abiotic source of Archean H₂O₂ and O₂, investigated the formation of ROS at the surface of the abraded quartz in anoxic conditions and suggested an essential but previously unappreciated role in the evolution of oxygenic photosynthesis on the early Earth. Because the core of this work is introducing this abiotic H₂O₂ producing mechanism to the exploration of the origin of oxygenic photosynthesis, we now provide a brief introduction of the production of ROS by mineral surfaces to better focus the theme of this paper (the scientific problem in this work).

The investigation of the formation of SBRs and ROS in anoxic conditions is the key observation from the interactions among abraded quartz, O₂ and water. Due to its serious harm to human health, quartz has aroused scientists' concern for a long time. The pathogenicity of silicosis is possibly caused by the radicals generated at the surface of the crushed quartz dusts. As being well reviewed in Schoonen et al. (2006), Fubini's group recognized surface-bound radicals and their reactivity. The presence of ≡Si•, ≡SiO• and ≡SiOO• on the surface of the abraded quartz was initially confirmed via EPR measurements⁷⁻⁹, and other forms of SBRs (such as ≡SiO₃•, ≡Si⁺O₂•, O₃•) were identified thereafter¹⁰⁻¹⁴. Previous studies suggested that these SBRs could readily react with H₂O and O₂ to form ROS¹⁵⁻¹⁷, thereby the ROS production of the abraded quartz decreases obviously with aging time in air¹⁸. **However, the formation of SBRs at surface of abraded minerals and the resultant ROS generation remained unclear.** The density of SBRs dominates the surface reactivity of abraded quartz¹⁹, but the precise content of different radical forms on the surface of the abraded quartz is uncertain¹⁸, which is tightly linked to the specific composition of the resultant ROS. Among the SBRs, the origin of SiOO• is also controversial. Giamello et al. (1990) inferred that the ≡SiOO• arose from reaction of broken ≡Si-O bonds with O₂ during mechanical grinding¹², but Murashov et al. (2003) and Balk et al. (2009) argued that the ≡SiOO• formed at the absence of O₂^{20,21}. In addition, knowledge gaps exist in the contribution of O₂ to the specific species, the kinetic of ROS release, and the variation of these species in abraded quartz-water reactions.

To elucidate the complex reactions between the quartz surface and water, we successfully performed strictly anoxic experiments and eliminated the shielding effect of pre-existing oxygen. Our results demonstrate that:

- (1) The ≡SiOO• could form at the surface of the abraded quartz at the absence of pre-existing atmospheric O₂ in mechanical grinding, which possibly contributes about 47.13% of H₂O₂ production,
- (2) The atmospheric O₂ involved into the mechanical grinding of quartz enhanced both the production of •OH and H₂O₂, while the dissolved O₂ in the water merely contributed to the increase of H₂O₂ production,

(3) The release of ROS and O₂ under anoxic conditions at the abraded quartz-water interfaces experienced an initial rapid rise and subsequent fluctuations, then ended gradually, in which most of the ROS and O₂ converted to H₂O₂, and

(4) Besides the •OH, the other two transient ROS (•O₂⁻ and ¹O₂) generated continuously in the dynamic abrasion of quartz.

This is a fundamental and essential work for the discussion about the surface reactivity of pulverized minerals in anoxic conditions (e.g., the moon, Mars and the early Earth with an anoxic atmosphere). Based on the accurate anoxic experimental data, we modelled time-course of H₂O₂ concentrations in local environments in the Archean. The calculation results indicated that abraded quartz in the anoxic Archean atmospheres not only could provide sufficient oxidants to fuel the respiration of the smallest cells (3 nM dissolved O₂), but it could have promoted the development of oxygen tolerance and spurred oxygenic photosynthesis. This abiotic H₂O₂ producing mechanism would have played an important role in the oxidative geochemistry of the Archean before the impact of the oxygenic photosynthesis.

Comment 2: The strong mechanical force induced by ball milling is also not well justified. This is the key point for the production of surface defects and ROS. The particle size and morphology of the quartz before/after ball milling were not well characterized. The relevance of such strong mechanical grinding (both intensity and time span) to natural environments was not justified. The calculations were mainly based on quartz surface area, but the H₂O₂ flux is heavily dependent on the intensity and extent of grinding.

Response: We break down our response below to address each of the 4 points within this comment.

(1) Justifying the use of ball milling – The reviewer is correct in that we ought to have provided a clearer statement for the purpose of the ball-milling experiment at 350 rpm. We performed ball-milling of quartz at 350 rpm in anoxic conditions to obtain fine powder with sufficient SBRs, which could produce clear signals so that we could discuss the formation mechanism of SBRs and ROS. Schoonen et al. (2006) stated that the stress-induced reactivity of quartz is dominated by contributions from the surface, and the density of SBRs on the abraded surface is not dependent on strain or grinding time. With the fine quartz powder, we verified the presence of SBRs (especially SiOO•) (**Fig. 1**) and illustrated how ROS was generated at the abraded quartz-water interface and showed the kinetics of ROS and O₂ release (**Fig. 2**). We have reworded the sentence (Lines 116 to 121).

(2) Characterizing particle size and morphology – We characterized the morphology and particle size before and after ball milling (**Fig. R1**). After grinding for 5 h, the raw quartz sands (0.25 ~ 0.6 mm) were crushed to fine powders (QGN-5h). The results of the particle-size measurements of QGN-5h show that the size of 97% of the particles is larger than 100 nm, and 70% of the particles has a diameter in the range of 1 to 20 μm (**Appendix A**). We have added this characterization into **Supplementary Information (Methods, Extended Data Fig. 1)**.

Fig. R1. The morphology and particle size before and after ball milling. (a) The photograph of quartz sands (0.25 ~ 0.6 mm); (b) The SEM image of the fine powder of quartz.

(3) Justifying intensity and time of grinding – The strong mechanical grinding (ball milling at 350 rpm for 5h) in experiments was different from the real abrasion of quartz in natural environments. To simulate the natural erosion of quartz on a sandy beach in a high-energy coastal environment, we used a pot-shaped tumbling barrel to grind quartz (**Methods, Extended Data Fig. 8**). Such an apparatus has been used to simulate the fluvial abrasion of small pebbles and sands in previous laboratory experiments^{22,23}. When the tumbling barrel was rolling at $1 \text{ m}\cdot\text{s}^{-1}$, mild collisions occurred among quartz sands (0.25 ~ 0.6 mm) and several small pebbles (1 ~ 2 cm) under the forces of gravity and the flowing water (**Fig. R2 a-b**). The mechanical process in the tumbling barrel involves interparticle grinding, including face grinding to a limited extent, but also with interparticle effects produced by short-distance rolling, sliding and movements within the sediment pack²³. Under the mechanical grinding, the size of quartz particles was gradually reduced and the initially clear water became cloudy. After 75 g of quartz was ground in 300 ml of water for 120 h, 10 ml of the suspension of quartz was sampled and the measured H_2O_2 concentration in the filtrate was $1.57 \text{ }\mu\text{M}$ (**Fig. R2 c**). Despite of the contribution of the atmospheric O_2 to the H_2O_2 production in the tumbling barrel experiments (as the results in **Extended Data Fig. 8** suggested), the appearance of H_2O_2 itself was enough to support the continuous generation of H_2O_2 in hydrodynamic environments of rivers and shores.

(4) Explaining the calculations – Based on the linear change of the specific surface area of quartz in the tumbling barrel, we calculated the fining rate in such a mild mechanical comminution condition. The results indicated that 129.75 m^2 of surface area can be produced by the abrasion of 1 g of quartz in 1 year (**Fig. R2 d**). Actually, the size distribution of quartz sands on the beach changes very little, because fine powders are carried by the currents to deep waters and deposited while newly produced coarse sands add to this zone during the erosion on the bed rock along the shore. Thus, the increase of the newly produced surface area in this water column maintained a linear trend. To show this clearly, we added some more details into **Supplementary Information (Methods, Supplementary Text, Extended Data Fig. 8 and Supplementary Video)**.

Fig. R2. The simulated physical erosion of quartz particles of a sandy beach. (a) Waves on a sandy beach. (b) Photograph of the collisions of quartz in the tumbling barrel; (c) Photograph of the cuvettes containing quartz slurry filtrate and LCV reagents; (d) The increasing rate of the specific surface area of quartz in such a mechanical process.

Comment 3: L25, L36: abstract should not contain references.

Response: We have removed the references from the abstract.

Comment 4: L67-69. This statement should be revised. The formation of Mn(III, IV) oxide is typically considered to involve the presence of oxygen. But here the authors are discussing an anoxic environment. Also, the Mn in Mn-catalase and WOC are in the reduced state, so the source of Mn(II) might be from direct dissolution of Mn(II)-containing minerals instead of redox reactions between Mn oxides and H₂O₂.

Response: The sentence "...inorganic minerals, such as a variety of manganese oxides" has been changed to "inorganic Mn complexes, such as manganese bicarbonate and a variety of MnO₂ minerals" (Lines 71-72). Despite the fact that oxygen in the Archean atmosphere was extremely low, a recent study²⁴ showed that Mn(II) could be oxidized photochemically.

Comment 5: L103-104. Reference needed.

Response: The following reference has now been cited in the text (Line 112).

Schrader, R., Wissing, R. & Kubsch, H. Zur Oberflächenchemie von mechanisch aktiviertem Quarz. *Z. Anorg. Allg. Chem.* 365, 191–198 (1969).

Comment 6: L121-123. Such peak broadening is more likely to be related to the reduction of particle size and decreasing crystallinity, a very common phenomenon for nanoparticles. Did the authors characterize the particle morphology and particle size before and after ball milling?

Response: We'd like to point out that after etching the amorphous surface of QGN-5h with HF, the peak became sharp-and-narrow in the obtained sample (QW) (**Fig. 1b**). Thus, the amorphous layer at the surface of quartz should be responsible for the subtle change in the XRD patterns. This concern is also mentioned in **Comment 2**, where we have added the characterization of the particle morphology and particle size before and after ball milling. Please see the **Response for Comment 2**.

Comment 7: The experiments with O₂ showed positive correlation between ROS production and O₂ concentration. The authors indicated that the anoxic grinding experiments were conducted in N₂. However, commercial ultrapure N₂ gas typically contains trace amount of O₂ (a few ppm), which might be inductive to the reactions. Did the authors measure the oxygen concentration of the gas?

Response: We have requested the gas supplier for their report of the compositions of high-purity N₂ gas (see **Appendix B**) and measured it by ourselves by gas chromatography, which consistently showed that the purity of N₂ is 99.999%, while the contents of O₂ and H₂O were below 2 and 3 ppm, respectively. In addition, the trace amount of O₂ and H₂O in the glovebox were monitored in real time (both were lower than 0.1 ppm) (**Fig. R3**), and the gas was continuously recycled by flowing through a column filled with copper catalyst particles and 5 Å molecular sieve to remove O₂ and H₂O, respectively. Thus, we are confident that there is almost no pre-existing O₂ in glovebox.

Fig. R3 The glovebox (a) used for our experiments, and the screenshot of the real-time monitoring values of the content of O₂ and H₂O (b) during the experiments.

Comment 8: The grinding experiments were conducted outside the glovebox with the jars filled with N₂. Did the authors ensure that the jars were well-sealed during grinding? Since the authors focused on the anoxic production of ROS during grinding, testing of oxygen level or potential leaking is critical.

Response: Thank you for your comment. Indeed, before we conducted the batch experiments, we had already run blanks to check the air tightness of the ball-milling jar. To ensure the fidelity of our experimental results, we checked the experimental set again and the results are shown in **Fig. R4**. After pumping a vacuum (the relative gas pressure in the jar was <3.9 mbar), the jar can keep this vacuum for >5 h. Thus, we are sure that there is no leakage when the jar is filled with N₂ (>0.1 MPa).

Fig. R4 Photograph of the vacuumed milling jar

Comment 9: The grinding experiments were conducted in a ball mill at 350 rpm. How does this translate to mechanical forces related to natural systems such as rives and tides? The ball milling also generated amorphous layers on quartz surface, which directly contributes to surface defects and production of ROS. Should we expect natural grinding processes by tides or waves to generate such intensity and kinetics of surface and structure transformation?

Response: This concern is referred in **Comment 2**, where we stated that we have added details of the experiments that simulate the natural abrasion of quartz on a sandy beach into **Supplementary Information (Methods, Supplementary Text, Extended Data Fig. 8)**.

Comments from Reviewer #3

Comment 1: The He et al., manuscript demonstrates laboratory-based abiotic generation of hydrogen peroxide (H_2O_2) due to quartz grain abrasion, which is proposed to be a potential electron donor and oxidant that exerted an evolutionary pressure for the emergence of oxygenic photosynthesis in the Archean eon. The 3.22 Ga Moodies group microbial mats are also proposed to have geochemical evidence that potentially supports the presence of hydrogen peroxide production via the quartz abrasion mechanism that would have promoted the eventual evolution of oxygenic photosynthesis. This is a very interesting study and mechanism for H_2O_2 production prior to the evolution of oxygenic photosynthesis. However, I think there needs to be a stronger link between the observed laboratory mechanism and natural conditions. There are several issues that should be addressed to warrant publication in Nature Communications.

Response: Thank you for your positive comments. As you will read below, we have now provided more text devoted to linking the experiments to natural conditions – both in the main text and supplemental information.

Comment 2: It is not clear if the ball-milling of quartz in the lab is a realistic representation of natural quartz abrasion caused in turbulent waterways or coastlines. How do the abrasive forces of the experiment compare to sandy riverbeds, deltas, or wave impact zones?

Response: Thank you for your comment. This is a common concern on the relevance between the ball-milling of quartz (e.g., at the speed of 350 rpm) in the lab and natural quartz abrasion. We conducted the experiments in the tumbling barrel to simulate of natural abrasion of quartz on a sandy beach (the velocity of the wave is $1 \text{ m}\cdot\text{s}^{-1}$). We apologize that we did not provide a complete description of these key details in our initial submission. Our simulation results show that the mild abrasion could produce quartz surface at a considerable rate of $129.75 \text{ m}^2\cdot\text{g}^{-1}\cdot\text{yr}^{-1}$, which is sufficient to create a locally oxidized environment in the shallow waters in the Archean (**Extended Data Fig. 10d**). To clarify what we did, we have now added some more details into **Supplementary Information (Methods, Supplementary Text, Extended Data Fig. 8)**. As **Reviewer#2** also has a similar concern, please find our corresponding **Response to Comment 2** on this point.

Comment 3: Turbulent events/conditions that are proposed to result in H_2O_2 production are often episodic. Any reduction in H_2O_2 production would result in rapid dilution of H_2O_2 to coastal waters. How would temporal variation impact H_2O_2 concentrations and impact on microbial communities?

Response: On the Archean delta/shore, the shallow water column, including the sand deposit, is constantly agitated by waves and tides (**Extended Data Fig. 8**), in which the quartz sand is continuously abraded. We calculated the H_2O_2 production in such a column of 1.63 L within a depth of 0.2 m to show the *in-situ* flux of H_2O_2 in the coastal water. In this simple model, the abrasion of quartz by waves/tides serves as a sustainable H_2O_2 source. Of course, the temporal variation in the hydrodynamic conditions may result in the dilution of H_2O_2 . For example, a storm may have carried substantial Fe(II) from the ferruginous Archean ocean to the near-shore waters where mechanochemical H_2O_2 was produced. Micromolar concentrations of Fe(II) in the current might react with hydrogen peroxide to produce hydroxyl radicals, which causes cellular damage²⁵⁻²⁷. With the regulation of catalases, the H_2O_2 concentration may be depressed to a low level to protect the ancestral cyanobacteria from Fe(II) toxicity. When the H_2O_2 was removed from the water column and the oxygenic photosynthesis was suppressed by reductants, some ancestral cyanobacteria in microbial mats could switch to anoxygenic photosynthesis²⁸⁻³⁰. However, it is noteworthy that the H_2O_2 concentration could be regained to micromolar levels within several days (**Extended Data Fig. 10d**). Thus,

the *in-situ* H₂O₂ production on an Archean delta/shore could maintain at least locally weak oxic conditions for the early microbial communities.

Comment 4: The study indicates that H₂O₂ production by quartz abrasion is greater under oxic conditions. How does H₂O₂ production in the experiment compare with H₂O₂ levels in modern day oxic coastal waters, deltas, or turbulent rivers? How does H₂O₂ production in the lab compare with H₂O₂ production in the Yellow river in the quartz grain-mixing interface? Modern microbial communities of today are more extensive than in the Archean and would certainly impact H₂O₂ levels to a greater extent than in the Archean, this could be discussed when comparing H₂O₂ production in modern day natural systems.

Response: It's a pity that we have not sampled the modern waters with strong hydrodynamic processes. Field measurements suggest that the concentration of H₂O₂ in the upper layers of the ocean is on the order of 10⁻⁷ M^{31,32}, and the H₂O₂ concentration decreases from coastal waters to the waters in the open ocean³². Besides sunlight initiated redox reactions involving organic compounds, O₂ and trace metals³³, mechanical grinding of quartz sands along the shore would also result in H₂O₂ in modern seawater. With a relatively high concentration of mechanochemical H₂O₂ diffusing to the near-shore waters, the H₂O₂ concentration in the surface waters should have at least been on the order of 10⁻⁶ M. However, the catalytic decomposition of H₂O₂ mediated by enzymes, transition metals halides and other trace constituents of seawater collectively lead to a short life time of H₂O₂. Thus, the actual H₂O₂ concentration is as low as 10⁻⁷ M.

To model the H₂O₂ flux from a river in the Archean, we took the Yellow River as a reference because its catchment area is only covered by a low fraction of vegetation. We assumed that a substantial fraction of the H₂O₂ generated at the suspended quartz surface is retained in the river water and carried to the delta.

In the modern ocean, marine plankton and bacteria contain peroxidases and catalases and chloroplasts contain a peroxidase mediated H₂O₂ scavenging system. When the H₂O₂ concentration rises to the point that cause toxicity to cells, these microorganisms protect themselves by adjusting the levels of these enzymes. In addition, cell lysis produces a substantial number of cell-free enzymes in seawater³⁴. The ubiquitous presence of H₂O₂ scavenging enzymes in modern surface waters could play an important role in regulating H₂O₂ concentrations to low levels.

Comment 5: How would primitive tectonics taking place at 3.2 Ga impact H₂O₂ production? Could enhanced tectonic activity enhance H₂O₂ production?

Response: This is a great question, but one that we do not have a comprehensive answer to. As the reviewer will be aware, there is significant controversy over the type of plate tectonics that existed in the Archean. What we can state is that there was likely significant emergent continental crust at that time given remnants of the Kaapvaal Craton in South Africa and the Pilbara Craton of Western Australia^{35,36}. Indeed, much of the 3.22 Ga Moodies Group is believed to represent coastal sediment that drained granite-greenstone terrains of the Kaapvaal Craton. Moreover, sediments from the Moodies Group were dominated by tonalite, felsic volcanic rock, komatiite-basalt, and granite³⁷, suggesting that significant quartz-rich sands were transported to the coastal waters of the Kaapvaal Craton. Thus, continental weathering should have contributed to the increase of the H₂O₂ flux to the delta.

We have now added this point into the sentence in the main text (lines 330-332): "As more quartz-rich rocks (such as tonalite-trondhjemite-granodiorite) emerged in the elevating continental crust in the Archean (**Fig. 4b**), the quartz-rich rocks experienced aggressive erosion during the fluvial transport to deltas and shores".

Comment 6: Lines 22 to 24 is an awkward sentence, consider changing to something like: “Anoxygenic predecessors used a variety of reductants, such as H₂, H₂S or Fe(II), but the manner in which photosynthesis transitioned to water as the electron donor remains unclear.”

Response: Thank you for your suggestion. The sentence has been changed as advised (Lines 22-24).

Comment 7: Lines 24 to 26 could be clearer, consider changing to something like: “An intriguing hypothesis proposes that hydrogen peroxide (H₂O₂) once acted as the electron donor as an intermediate stage prior to the evolution of oxygenic photosynthesis, but its abundance during the Archean would have been limited.”

Response: Thank you for your suggestion. The sentence has been changed as advised (Lines 26-27).

Comment 8: Line 27: insert the word “production” after “H₂O₂”

Response: Thank you for the suggestion. We have inserted the word “production” after “H₂O₂” (Line 28).

Comment 9: Lines 34 to 37: the final sentence of the abstract is too passive. There should be more evidence stated here in the abstract to connect these microbial mats to the use of H₂O₂ as an electron donor (redox evidence described in the discussion).

Response: This sentence has now been reworded (Lines 36-40).

Comment 10: The main text needs to be separated into subsections with subheadings, including a conclusions section at the end. This will provide more structure to the manuscript.

Response: Thank you for the suggestion. Five subheadings have now been added to the text as follows:

1. The SBRs-induced generation of ROS at the abraded quartz-water interface
2. Continuous generation of ROS in hydrodynamic environments with abrasions of quartz
3. The H₂O₂ flux and the resultant redox evolution in the Archean
4. The H₂O₂-driving evolution of photosynthesis
5. Conclusions

Comment 11: Lines 40 to 41: change to “...biological innovation that allowed water to be used as an electron source and dioxygen gas...”

Response: This sentence has been changed to “...biological innovation that allowed water to be used as an electron source and dioxygen gas...” (Lines 43-44).

Comment 12: Line 106: change to “...the Great Oxidation Event (GOE)...”

Response: We apologize for the error. It has been changed to “the Great Oxidation Event” (Line 122).

Comment 13: Line 107: “Farquhar et al., 2000” should be cited here as well, and the range of “2.45 Ga to 2.3 Ga” is more appropriate than one time point at 2.45 Ga.

Atmospheric Influence of Earth's Earliest Sulfur Cycle

James Farquhar*, Huiming Bao, Mark Thiemens

Science 04 Aug 2000:

Vol. 289, Issue 5480, pp. 756-758

DOI: 10.1126/science.289.5480.756

Response: Thank you for this suggestion. This reference has been cited in the text (Line 122). The time range for the GOE has also been modified.

Comment 14: Line 187: why is “(REF.)” needed here in parentheses? Could you just add reference 31 at the end of the sentence?

Response: We have deleted “(REF.)” (Line 197).

Comment 15: Line 314: delete “the” before “early oxygenic photosynthesis.”

Response: “the” has been deleted (Line 342).

Reference

1. Fubini, B. & Areán, C. O. Chemical aspects of the toxicity of inhaled mineral dusts. *Chem. Soc. Rev.* **28**, 373–381 (1999).
2. Fubini, B. & Hubbard, A. Reactive oxygen species (ROS) and reactive nitrogen species (RNS) generation by silica in inflammation and fibrosis. *Free Radic. Biol. Med.* **34**, 1507–1516 (2003).
3. Cohn, C. A., Laffers, R., Simon, S. R., O’Riordan, T. & Schoonen, M. A. Role of pyrite in formation of hydroxyl radicals in coal: possible implications for human health. *Part. Fibre. Toxicol.* **3**, 16 (2006).
4. Harrington, A. D., Tsirka, S. E. & Schoonen, M. A. Quantification of particle-induced inflammatory stress response: a novel approach for toxicity testing of earth materials. *Geochem. Trans.* **13**, 4 (2012).
5. Xu, J., Sahai, N., Eggleston, C. M. & Schoonen, M. A. A. Reactive oxygen species at the oxide/water interface: Formation mechanisms and implications for prebiotic chemistry and the origin of life. *Earth Planet. Sci. Lett.* **363**, 156–167 (2013).
6. Kaur, J. & Schoonen, M. A. Non-linear hydroxyl radical formation rate in dispersions containing mixtures of pyrite and chalcopyrite particles. *Geochim. Cosmochim. Acta* **206**, 364–378 (2017).
7. Schrader, R., Wissing, R. & Kubsch, H. Zur Oberflächenchemie von mechanisch aktiviertem Quarz. *Z. Anorg. Allg. Chem.* **365**, 191–198 (1969).
8. Hochstrasser, G. & Antonini, J. F. Surface states of pristine silica surfaces: I. ESR studies of Es’ dangling bonds and of CO₂-adsorbed radicals. *Surf. Sci.* **32**, 644–664 (1972).
9. Ratdzig, V. A. & Bystrikov, A. V. ESR study of chemically active-centers on the surface of quartz. *Kinet. Catal.* **19**, 563–568 (1978).
10. Fubini, B., Giamello, E., Pugliese, L. & Volante, M. Mechanically induced defects in quartz and their impact on pathogenicity. *Solid State Ion* **32–33**, 334–343 (1989).
11. Fubini, B., Bolis, V. & Giamello, E. The surface chemistry of crushed quartz dust in relation to its pathogenicity. *Inorg. Chim. Acta* **5** (1987).
12. Giamello, E., Fubini, B., Volante, M. & Costa, D. Surface oxygen radicals originating via redox reactions during the mechanical activation of crystalline SiO₂ in hydrogen peroxide. *Colloids Surf.* **45**, 155–165 (1990).
13. Zavyalov, S. A. & Streletskii, A. N. Role of O₂ radical anions in the formation and emission of molecules of singlet oxygen from a surface of freshly crushed quartz. *Kinet. Catal.* **26**, 1005–1008 (1985).
14. Costa, D., Fubini, B., Giamello, E. & Volante, M. A novel type of active site at the surface of crystalline SiO₂ (α-quartz)

- and its possible impact on pathogenicity. *Can. J. Chem.* **69**, 1427–1434 (1991).
15. Fenoglio, I. *et al.* The role of mechanochemistry in the pulmonary toxicity caused by particulate minerals. *J. Mater. Synth. Proces.* **8**, 145–153 (2000).
 16. Vallyathan, V. *et al.* Freshly fractured quartz inhalation leads to enhanced lung injury and inflammation. Potential role of free radicals. *Am. J. Respir. Crit. Care Med.* **152**, 1003–1009 (1995).
 17. Shi, X., Dalai, N. S. & Vallyathan, V. ESR evidence for the hydroxyl radical formation in aqueous suspension of quartz particles and its possible significance to lipid peroxidation in silicosis. *J. Toxicol. Environ. Health* **25**, 237–245 (1988).
 18. Fubini, B., Giamello, E., Volante, M. & Bolis, V. Chemical functionalities at the silica surface determining its reactivity when inhaled. Formation and reactivity of surface radicals. *Toxicol. Ind. Health* **6**, 571–598 (1990).
 19. Schoonen, M. A. A. *et al.* Mineral-induced formation of reactive oxygen species. *Rev. Mineral Geochem.* **64**, 179–221 (2006).
 20. Murashov, V. Ab initio cluster calculations of silica surface sites. *J. Mol. Struct.* **650**, 141–157 (2003).
 21. Balk, M. *et al.* Oxidation of water to hydrogen peroxide at the rock–water interface due to stress-activated electric currents in rocks. *Earth Planet. Sci. Lett.* **283**, 87–92 (2009).
 22. Kuenen, P. H. Experimental abrasion 3. Fluvial action on sand. *Am. J. Sci.* **257**, 172–190 (1959).
 23. Lewin, J. & Brewer, P. A. Laboratory simulation of clast abrasion. *Earth Surf. Process. Landf.* **27**, 145–164 (2002).
 24. Liu, W. *et al.* Anoxic photogeochemical oxidation of manganese carbonate yields manganese oxide. *Proc. Natl. Acad. Sci. U.S.A.* **117**, 22698–22704 (2020).
 25. Imlay, J. A. Pathways of oxidative damage. *Annu. Rev. Microbiol.* **57**, 395–418 (2003).
 26. Imlay, J. A. Cellular defenses against superoxide and hydrogen peroxide. *Annu. Rev. Biochem.* **77**, 755–776 (2008).
 27. Szeinbaum, N., Toporek, Y. J., Reinhard, C. T. & Glass, J. B. Microbial helpers allow cyanobacteria to thrive in ferruginous waters. *Geobiology* **0**, 1–11 (2021).
 28. Padan, E. Facultative anoxygenic photosynthesis in cyanobacteria. *Annu. Rev. Plant. Biol.* **30**, 27–40 (1979).
 29. Dick, G. J., Grim, S. L. & Klatt, J. M. Controls on O₂ production in cyanobacterial mats and implications for Earth's oxygenation. *Annu. Rev. Earth Planet. Sci.* **46**, 123–147 (2018).
 30. Liu, D. *et al.* *Synechococcus* sp. strain PCC7002 uses sulfide: quinone oxidoreductase to detoxify exogenous sulfide and to convert endogenous sulfide to cellular sulfane sulfur. *Mbio* **11**, (2020).
 31. Zika, R. G., Moffett, J. W., Petasne, R. G., Cooper, W. J. & Saltzman, E. S. Spatial and temporal variations of hydrogen

- peroxide in Gulf of Mexico waters. *Geochim. Cosmochim. Acta* **49**, 1173–1184 (1985).
32. Petasne, R. G. & Zika, R. G. Hydrogen peroxide lifetimes in south Florida coastal and offshore waters. *Mar. Chem.* **56**, 215–225 (1997).
 33. Zika, R. G. Chapter 10 Marine Organic Photochemistry. in *Elsevier Oceanography Series* (eds. Duursma, E. K. & Dawson, R.) vol. 31 299–325 (Elsevier, 1981).
 34. Kim, J. & ZoBell, C. E. Occurrence and activities of cell-free enzymes in oceanic environments. in *Effect of the Ocean Environment on Microbial Activities* (eds. Colwell, R. R. & Mortia, R. Y.) 368–385 (University Park Press, 1974).
 35. Jahn, B.-M. & Condie, K. C. Evolution of the Kaapvaal Craton as viewed from geochemical and Sm-Nd isotopic analyses of intracratonic pelites. *Geochim. Cosmochim. Acta* **59**, 2239–2258 (1995).
 36. Kranendonk, M. J. V., Smithies, R. H., Hickman, A. H. & Champion, D. C. Review: secular tectonic evolution of Archean continental crust: interplay between horizontal and vertical processes in the formation of the Pilbara Craton, Australia. *Terra Nova* **19**, 1–38 (2007).
 37. Hessler, A. M. & Lowe, D. R. Weathering and sediment generation in the Archean: An integrated study of the evolution of siliciclastic sedimentary rocks of the 3.2Ga Moodies Group, Barberton Greenstone Belt, South Africa. *Precambrian Res.* **151**, 185–210 (2006).

Appendix A

The test report of the particle size of quartz sample (QGN-5h)

JL-1177型激光粒度测试仪测试报告

试样名称:	D3: 0.188 μm	D6: 0.244 μm
试样编号:	D10: 0.320 μm	D16: 0.455 μm
介质名称:	D25: 0.725 μm	D50: 1.899 μm
操作人员:	D75: 6.342 μm	D84: 9.556 μm
浓度: 66	D90: 12.189 μm	D97: 17.821 μm
表面积/体积: 6.7256m ² /cm ³	D98: 19.231 μm	Mode: 体积自由分布
体积平均粒径[D4, 3]: 4.292 μm		样品折射率: 1.51 + 0.010i
测试日期: 2021-05-19 14:59:19	文件名:	介质折射率: 1.33 + 0.000i

粒径 (μm)	积分 (%)	微分 (%)	粒径 (μm)	积分 (%)	微分 (%)	粒径 (μm)	积分 (%)	微分 (%)
0.020	0.000	0.000	1.476	43.331	3.926	54.27	100.000	0.000
0.035	0.000	0.000	1.687	46.911	3.580	62.02	100.000	0.000
0.050	0.002	0.002	1.928	50.422	3.510	70.88	100.000	0.000
0.060	0.017	0.015	2.204	53.676	3.255	81.00	100.000	0.000
0.069	0.037	0.020	2.518	56.782	3.106	92.57	100.000	0.000
0.078	0.065	0.028	2.878	59.564	2.781	105.8	100.000	0.000
0.089	0.107	0.042	3.289	62.149	2.586	120.9	100.000	0.000
0.102	0.176	0.069	3.759	64.881	2.731	138.2	100.000	0.000
0.117	0.279	0.104	4.296	67.444	2.564	157.9	100.000	0.000
0.134	0.412	0.133	4.909	69.924	2.480	180.5	100.000	0.000
0.153	0.591	0.179	5.610	72.573	2.649	206.2	100.000	0.000
0.174	0.835	0.244	6.411	75.229	2.656	235.7	100.000	0.000
0.199	1.144	0.309	7.327	78.101	2.872	269.3	100.000	0.000
0.228	1.514	0.370	8.373	81.080	2.978	307.8	100.000	0.000
0.260	1.946	0.432	9.569	84.032	2.952	351.8	100.000	0.000
0.297	2.439	0.493	10.94	87.227	3.195	402.0	100.000	0.000
0.340	2.993	0.554	12.50	90.684	3.458	459.4	100.000	0.000
0.389	3.607	0.615	14.28	93.674	2.990	525.0	100.000	0.000
0.444	4.282	0.676	16.32	95.830	2.156	600.0	100.000	0.000
0.507	5.018	0.737	18.65	97.649	1.818	712.6	100.000	0.000
0.580	5.813	0.798	21.32	99.268	1.619	846.3	100.000	0.000
0.663	6.667	0.859	24.36	100.000	0.732	1005	100.000	0.000
0.757	7.576	0.920	27.84	100.000	0.000	1194	100.000	0.000
0.866	8.550	0.981	31.82	100.000	0.000	1415	100.000	0.000
0.989	9.589	1.042	36.36	100.000	0.000	1684	100.000	0.000
1.130	10.691	1.103	41.56	100.000	0.000	2000	100.000	0.000
1.292	11.848	1.164	47.49	100.000	0.000			

Appendix B

The test report of the high-purity N₂ gas used for the experiments

广州市粤佳气体有限公司

GUANGZHOU YIGAS GASES CO., LTD

报告编号 NO: 20210428001

气体检验报告

TEST REPORT

气体名称: Gas name	高纯氮	钢瓶规格: Packaging	T40L
生产数量: The number of production	60 瓶	检验日期: Date of test	2021/0428
检验标准: Test standard	GB/T 8979-2008	检验仪器: Testing equipment	气相色谱仪;水分仪;氧份仪
检验规则: Test rule	全检	检验数量: The number of test	60 瓶
检验结果 (Result Of Test)			
检测项目	指标	检测数据	判定
高纯氮 (N ₂) (%)	≥99.999	>99.999	√
氧气 (O ₂) (ppm)	≤3	≤2	√
氢气 (H ₂) (ppm)	≤1	≤0.2	√
一氧化碳 (CO) (ppm)	≤1	<1	√
二氧化碳 (CO ₂) (ppm)	≤1	<1	√
甲烷 (CH ₄) (ppm)	≤1	≤0.2	√
水分 (H ₂ O) (ppm)	≤3	≤3	√
(以下空白)			
检验结论 Test Conclusion 出厂产品符合 GB/T 8979-2008 高纯氮要求 			
备注: Note			

检验员: 何庆荣

地址: 广州市增城新塘镇大王岗工业区
电话: 020-82797785

邮编: 511340
网址: www.51qiti.com

REVIEWERS' COMMENTS

Reviewer #2 (Remarks to the Author):

The authors have addressed my comments and the manuscript is improved in clarity and presentation.

Reviewer #3 (Remarks to the Author):

Thank you to the authors for providing the revised manuscript. Overall, the authors have strengthened the manuscript and adequately addressed all the reviewer comments to warrant publication, particularly with respect to connecting lab results to natural conditions. There are a few small remaining issues that could still be addressed to improve the manuscript.

Follow up to comment 4 from Reviewer 3 – please provide a reference for the concentration of H₂O₂ in coastal waters to compare to the open ocean concentration of 10⁻⁷ M described in the response comment by the authors. It appears that the references provided in the comment response (Zika et al., 1985; Petasne and Zika, 1997) can be used to report H₂O₂ levels for coastal and offshore waters.

Line 36 of the revised manuscript: delete the word "The" at the beginning of the sentence and change the beginning of the sentence to: "Early microbial mats intercalated with sandy layers, as observed in the 3.22 billion year old Moodies Group in South Africa, might have performed..."

Line 343 of the revised manuscript: change the end of the sentence to "...necessarily tied to early oxygenic photosynthesis."